# A Location-Aware Resource Optimization for Maximizing Throughput of Emergency Outdoor–Indoor UAV Communication with FSO/RF

**DOI:** 10.3390/s23052541

**Published:** 2023-02-24

**Authors:** Zinan Guo, Wei Gao, Haijun Ye, Guofeng Wang

**Affiliations:** 1State Key Laboratory of Networking and Switching Technology, Beijing University of Posts and Telecommunications, Beijing 100876, China; 2The China Electronics Science Research Institute, Beijing 100041, China; 3The School of Information and Communication Engineering, Beijing University of Posts and Telecommunications, Beijing 100876, China

**Keywords:** UAV, FSO/RF, throughput, outdoor–indoor transmission, fairness

## Abstract

In emergency communication scenarios, unmanned aerial vehicles (UAVs) can be used as an air relay to provide higher-quality communication for indoor users. When bandwidth resources are scarce, the use of free space optics (FSO) technology will greatly improve the resource utilization of the communication system. Therefore, we introduce FSO technology into the backhaul link of outdoor communication, and use free space optical/radio frequency (FSO/RF) technology to realize the access link of outdoor indoor communication. The deployment location of UAVs will affect not only the through wall loss of outdoor–indoor communication but also the quality of FSO communication, and, therefore, it needs to be optimized. In addition, by optimizing the power and bandwidth allocation of UAVs, we realize the efficient utilization of resources and improve the system throughput on the premise of considering information causality constraints and user fairness. The simulation results show that, by optimizing the location and power bandwidth allocation of UAVs, the system throughput is maximized, and the throughput between each user is fair.

## 1. Introduction

With the advantages of high flexibility and low deployment cost, unmanned aerial vehicles (UAVs) are being increasingly and more widely used, having attracted the attention of various industries. From 2015 to 2025, the global UAV market will grow from USD 60 billion to nearly USD 90 billion [1]. In January 2017, the International Telecommunication Union launched a standard research project for civil UAV Communication Services [2], which marks the gradual expansion of UAV research in the field of communication to civil use. According to research by Huawei [3], 70% of the traffic will occur indoors in the 5G era. This finding reflects the expectation that a large number of users will be within indoor environments. In an emergency communication scenario, users of indoor environments may face secondary disasters, such as fire and house collapse, with a much higher risk coefficient than for outdoor users. Therefore, in terms of emergency communication, guaranteed communication should be provided for ground outdoor users, but necessary communication should also be provided for indoor users [4], so as to effectively protect people’s lives and safety. The larger the throughput of the communication system, the higher the transmission rate, which can better ensure the real-time high-speed communication of users [5].

FSO communication does not use optical fiber and other guided wave media. It directly uses laser for signal transmission in the atmosphere and can carry out high-speed two-way transmission of voice, data, television, and multimedia images [6]. FSO has the advantages of being applicable to any communication protocol and highly economical in addition to having large transmission capacity and high data transmission rates [7]. It is thus a major research hotspot globally. In outdoor communication, UAVs can use FSO technology to increase system throughput [8].

Indoor users and outdoor base stations utilize an outdoor–indoor link model [4]. Outdoor users and outdoor base stations use an air-to-ground (ATG) link model [9]. In terms of signal propagation loss, there are line of sight (LOS) and non-line of sight (NLOS) components in the ATG link established by the signal arriving at the outdoor user, and the transmission loss comprises only outdoor transmission loss. The NLOS link must be established when the signal reaches the indoor user. The transmission loss includes outdoor transmission loss, signal through wall loss, and indoor transmission loss. In addition, the elevation angle of UAV antenna not only affects the coverage area but also affects the signal through wall loss [10].

Wenzheng Xu et al. in [11] studied the deployment of multiple UAVs in providing communication services for emergency areas. An approximation algorithm was proposed that has greatly improved system throughput. Moreover, the solution effect of the proposed algorithm was found to be 12% higher than that of the existing algorithms. N. Gupta et al. in [12] solved the problem of UAV deployment to maximize the sum rate of the system. An iterative scheme for obtaining the optimal trajectory of UAV was proposed. In the proposed scheme, the optimal position of each time slot is calculated in turn, and the final position then obtained using a greedy method. The simulation results showed that the performance of the proposed scheme is 16.5% higher than the benchmark scheme on average. Jian Cui et al. in [13] optimized the main signal transmission path model by studying the system stability of UAV when establishing a remote relay link for indoor users. The joint optimization of UAV deployment location and bandwidth allocation ensured the minimal outage probability of the system. Z. Guo et al. in [5] used a model in the literature [13] to further optimize bandwidth and deployment location toward maximizing the throughput of the outdoor–indoor communication system. Due to the different link models at both ends of UAV relay, the system had the problem of asymmetry. In order to solve the practical problems in asymmetric double-hop relay system, the authors mainly focused on buffer auxiliary relay. In the literature [14], it has been shown that buffering can be used to improve system performance under various mixed free space optical/radio frequency (FSO/RF) link conditions. In [15], the buffer assisted cooperation protocol was optimized based on FSO, which demonstrated that the increase in buffer freedom of FSO assisted relay occurs at the cost of increasing delay. J.-H Lee et al. in [16] studied a two-hop hybrid FSO/RF UAV relay system (i.e., use of FSO and RF links for source to relay and relay to target links, respectively). On the premise of fully considering the unbalanced transmission rate of RF and FSO links, the trajectory of UAV was optimized to obtain the maximum data throughput for a ground user terminal.

Under the premise of information causality constraints, we jointly optimize the deployment location and power bandwidth allocation of a UAV to maximize the throughput of the UAV relay communication system. We propose a joint location and resource allocation optimization (JLRO) algorithm to ensure the fairness of users. In other words, all users have the same communication rate. The main contributions of this paper are as follows:We consider the emergency communication scenario, wherein UAV provides remote communication services for indoor users through the two-hop link model. In this model, the FSO-based outdoor transmission model is considered for the first hop channel, and the outdoor–indoor RF transmission model is considered for the second-hop channel. We construct a multidimensional nonconvex mathematical model of the UAV’s three-dimensional location and power bandwidth allocation. In this model, the user fairness principle and information causality constraint are considered.Based on the three-dimensional deployment location and power bandwidth allocation of the UAV, the JLRO algorithm is designed. Firstly, we decompose the complex original problem into two subproblems: location optimization and power bandwidth allocation. Then, an iterative algorithm based on block coordinate descent is designed for overall optimization.We use successive convex approximation (SCA) theory to solve two subproblems of the original problem. The simulation results show that the JLRO algorithm has good convergence and can maximize the throughput of the communication system and ensure fairness for all users.

The rest of the paper is organized as follows. In Section 2, a maximize throughput optimization problem for UAV relay networks based on FSO/RF is presented and formulated. In Section 3, the original problem is decomposed into two subproblems of location optimization and resource allocation, which are, respectively, solved, and the JLRO algorithm is then proposed. Section 4 and Section 5 present the simulation results and conclusions for this paper, respectively.

## 2. System Model and Problem Formulation

In the emergency communication scenario, indoor users should maintain real-time communication with outdoor base stations or command centers. This paper focuses on the throughput of downlink. The signal is transmitted from the outdoor base station to the indoor user in two steps. The first step is to transmit the signal from the outdoor base station to the UAV. The second step is to transmit the signal from UAV to indoor users. The first step is to transmit the signal from the outdoor base station to the UAV with consideration of FSO communication. The second step is to transmit the signal from UAV to indoor users with consideration of outdoor–indoor RF communication. The signal transmission process is shown in Figure 1a. As shown in Figure 1a, the UAV acts as an air relay to transmit the signal sent by the outdoor base station to the indoor user. In Figure 1b, *S* represents an outdoor base station (source node). *D* represents indoor user (destination node). $D_{i}$ represents the *i*-th indoor user. *R* stands for UAV (air relay node). $d_{S,R}$ represents the distance from the outdoor base station to the UAV. $d_{D_{i},R}$ represents the outdoor transmission distance from the UAV to the wall corresponding to the *i*-th indoor user. $d_{i}$ represents the horizontal distance between the *i*-th indoor user and the wall. $\theta_{i}$ represents the elevation angle of signal passing through the wall when UAV communicates with the ith indoor user. The coordinates of *R* are $\left(x_{U},y_{U},z_{U}\right)$. The coordinates of *S* are $\left(x_{S},y_{S},z_{S}\right)$. The coordinates of $D_{i}$ are $\left(x_{D_{i}},y_{D_{i}},z_{D_{i}}\right)$. The length, width, and height of the building are $x_{B}$, $y_{B}$, and $z_{B}$, respectively. Clearly, $x_{D_{i}}\leq x_{B}$, $y_{D_{i}}\leq y_{B}$ and $z_{D_{i}}\leq z_{B}$. The number of indoor users is *n*.

The symbols commonly used in the article are shown in Table 1.

### 2.1. The First Step

In an emergency communication scenario, there may be a long distance between available outdoor base stations and destinations. The features of long transmission distance, fast installation, and no license make FSO very suitable for short-term deployment. Therefore, we use FSO technology in the first step of transmission.

The FSO data transmission model we use is as follows:
(1a)$$\begin{array}{cc} \; & C=\frac{P_{s}\eta \phi 10^{-\frac{L_{a}}{10}}10^{-\frac{L_{h}}{10}}A_{R}}{A_{B}E_{p}N_{b}}\left[b/s\right], \end{array}$$
(1b)$$\begin{array}{cc} \; & E_{p}=\frac{h_{p}c}{\lambda }, \end{array}$$
(1c)$$\begin{array}{cc} \; & A_{R}=\pi r^{2}, \end{array}$$
(1d)$$\begin{array}{cc} \; & A_{B}=\pi {\left(\psi d_{S,R}\times 0.5\right)}^{2}=0.25\pi \psi^{2}d_{S,R}^{2}, \end{array}$$
(1e)$$\begin{array}{cc} \; & L_{a}=\frac{17}{V}{\left(\frac{\lambda }{550nm}\right)}^{-q}, \end{array}$$
(1f)$$\begin{array}{cc} \; & d_{S,R}=\sqrt{{\left(x_{U}-x_{S}\right)}^{2}+{\left(y_{U}-y_{S}\right)}^{2}+{\left(z_{U}-z_{S}\right)}^{2}} \end{array}$$
where $P_{s}$ is the transmission power of outdoor base station; η is the optical efficiency of the outdoor base station transmitting equipment; ϕ is the optical efficiency of UAV receiving equipment; $E_{p}$ represents photon energy; $h_{p}$ represents Planck’s constant; λ represents the carrier wavelength of the laser; $N_{b}$ represents the sensitivity of the UAV receiving equipment in the number of photons/bit; $L_{h}$ represents the higher pointing losses in dB; *r* represents the radius of the aperture of the UAV receiving equipment in *m*; the unit of $d_{S,R}$ is Km; ψ represents the transmitting divergence angle; $L_{a}$ represents the atmospheric attenuation caused by rain, fog, cloud, and other factors in dB/Km; *V* represents the visibility in Km; *q* represents the size distribution of scattered particles under different weather conditions. The functional relationship between *q* and *V* is:(2)$$q=\left\{\begin{array}{ccc} \; & 1.6, & V>50\;\mathrm{km}\\ \; & 1.3, & 6\;\mathrm{km}<V<50\;\mathrm{km}\\ \; & 0.16V+0.34, & 1\;\mathrm{km}<V<6\;\mathrm{km}\\ \; & V-0.5, & 0.5\;\mathrm{km}<V<1\;\mathrm{km}\\ \; & 0, & V<0.5\;\mathrm{km} \end{array}\right.$$

### 2.2. The Second Step

In this step, a signal is sent from the UAV. After transmitting a certain distance in the air, it needs to pass through the wall and then transmit over a certain distance indoors to reach the user terminal. Since the signal needs to pass through the wall, FSO will no longer be suitable for this process. The signal will go through three processes: outdoor transmission loss, through wall loss, and indoor transmission loss.

The total path loss during signal transmission from UAV to user terminal can be expressed as:
(3a)$$\begin{array}{cc} \; & PL_{i}=PL_{Fi}+PL_{Bi}+PL_{Ii}, \end{array}$$
(3b)$$\begin{array}{cc} \; & PL_{Fi}=20\log_{10}\left(\frac{4\pi fd_{D_{i},R}}{c}\right), \end{array}$$
(3c)$$\begin{array}{cc} \; & PL_{Bi}=\beta_{1}+\beta_{2}{\left(1-\cos\theta_{i}\right)}^{2}, \end{array}$$
(3d)$$\begin{array}{cc} \; & PL_{Ii}=\beta_{3}d_{i}, \end{array}$$
where
(4a)$$\begin{array}{cc} \; & d_{D_{i},R}=\sqrt{{\left(x_{U}-x_{B}\right)}^{2}+{\left(y_{U}-y_{i}\right)}^{2}+{\left(z_{U}-z_{i}\right)}^{2}}, \end{array}$$
(4b)$$\begin{array}{cc} \; & \cos\theta_{i}=\frac{x_{U}-x_{B}}{d_{D_{i},R}}, \end{array}$$
(4c)$$\begin{array}{cc} \; & d_{i}=x_{B}-x_{i}. \end{array}$$

In the Formula (3), $PL_{Fi}$ represents the path loss when the signal is transmitted outside the room. Generally speaking, the UAV deployment location is close to the building. Therefore, we use the free space path loss model to calculate the path loss. $PL_{Bi}$ represents the loss when the signal passes through the wall. The loss is greatly affected by wall material, thickness, color, and other factors. $PL_{Ii}$ represents the path loss of indoor signal transmission. This loss is greatly affected by environmental factors indoors. *c* is the speed of light. Under the ITU standard test environment [2], $\beta_{1}=14$, $\beta_{2}=15$, $\beta_{3}=0.5$. *f* is the carrier frequency, and its unit is Hz. The unit of path loss is dB. All distances are in meters.

The transmission rate of the second link for each user terminal can be expressed as:(5)$$\begin{array}{c} T_{i}=B_{i}\log_{2}\left(1+\frac{P_{i}10^{-\frac{PL_{i}}{10}}}{B_{i}N_{0}}\right). \end{array}$$

$P_{i}$ and $B_{i}$, respectively, represent the independent power and bandwidth allocated by the UAV to each user terminal. Bandwidth allocation avoids co frequency interference. $N_{0}$ is the noise power spectral density.

### 2.3. Problem Formulation

The goal of this research is to maximize the instantaneous transmission rate of the communication network by jointly optimizing the deployment location, bandwidth, and power allocation of UAV relay. This paper also considers user fairness and information causality constraints. The problem can be expressed as:
(6a)$$\begin{array}{cc} \; & \max_{B_{i},P_{i},x_{U},y_{U},z_{U}}\sum_{i=1}^{n}T_{i} \end{array}$$
(6b)$$\begin{array}{cc} s.t.\;\;\; & x_{B}\leq x_{U}\leq x_{S} \end{array}$$
(6c)$$\begin{array}{cc} \; & 0\leq y_{U}\leq y_{B} \end{array}$$
(6d)$$\begin{array}{cc} \; & 0\leq z_{U}\leq z_{B} \end{array}$$
(6e)$$\begin{array}{cc} \; & B_{i}\geq 0,P_{i}\geq 0,\forall i \end{array}$$
(6f)$$\begin{array}{cc} \; & \sum_{i=1}^{n}B_{i}\leq B_{max} \end{array}$$
(6g)$$\begin{array}{cc} \; & \sum_{i=1}^{n}P_{i}\leq P_{max} \end{array}$$
(6h)$$\begin{array}{cc} \; & T_{i}=T_{j}=\cdots =T_{n} \end{array}$$
(6i)$$\begin{array}{cc} \; & \sum_{i=1}^{n}T_{i}\leq C \end{array}$$

Through (6b)–(6d), the coordinates of the UAV are jointly limited. In order to reduce transmission loss, the UAV needs to hover on one side of the building. In this paper, the UAV hovers on the right side in Figure 1. Formula (6e) is the basic condition for normal communication, (6f) indicates that the sum of all sub bandwidths should not be higher than the total bandwidth of UAV, and (6g) indicates that the sum of all sub powers should not be higher than the total UAV power. Of course, under normal circumstances, only $\sum_{i=1}^{n}B_{i}=B_{max}$ and $\sum_{i=1}^{n}P_{i}=P_{max}$ can maximize the transmission rate of the second step. Formula (6h) represents fairness, namely that all users have the right to enjoy the same signal transmission rate. Formula (6i) represents the information causality constraint. This paper focuses on the downlink. The sum of the second step signal transmission rate should not exceed the first step signal transmission rate.

## 3. Algorithm Design

### 3.1. Problem Reformulation

It is hard to find an optimal solution to problem (6) due to the fact that it is a highly nonconvex issue. Firstly, in problem (6), both the information causality and fairness constraints are considered. Secondly, the differences between modeling of outdoor and indoor transmission also introduce additional difficulty in solving the problem (6). In the following, as shown in the Figure 2 for solving problem (6), we first change problem (6) into an equivalent form, and we then further divide the original problem into the UAV 3D location subproblem and the resource allocation subproblem. Finally, we propose an iteration-based algorithm as an alternative for solving the two previously mentioned subproblems through reaching convergency.

By introducing slack variable π to deal with the information causality constraint (6i), problem (6) can be equivalently transformed as:
(7a)$$\begin{array}{cc} \; & \max_{\pi ,B_{i},P_{i},x_{U},y_{U},z_{U}}\pi \end{array}$$
(7b)$$\begin{array}{cc} s.t.\;\;\; & \left(6b),\;(6c),\;(6d),\;(6e),\;(6f),\;(6g),\;(6h\right) \end{array}$$
(7c)$$\begin{array}{cc} \; & \pi \leq C \end{array}$$
(7d)$$\begin{array}{cc} \; & \pi \leq \sum_{i=1}^{n}T_{i} \end{array}$$

The fairness constraint (6h) in problem (7) can be handled by further introducing the lower bounded condition $\theta =\min\{T_{i},\forall i\}$; thus, problem (7) can be further equivalently transformed as:
(8a)$$\begin{array}{cc} \; & \max_{\theta ,\pi ,B_{i},P_{i},x_{U},y_{U},z_{U}}\pi \end{array}$$
(8b)$$\begin{array}{cc} s.t.\;\;\; & \left(6b),\;(6c),\;(6d),\;(6e),\;(6f),\;(6g),\;(7c\right) \end{array}$$
(8c)$$\begin{array}{cc} \; & \theta \leq T_{i},\forall i \end{array}$$
(8d)$$\begin{array}{cc} \; & \pi \leq n\theta \end{array}$$

Then, based on this equivalent problem (8) we will decouple it into two subproblems, i.e., UAV 3D location subproblem and resource allocation subproblem. Finally, we propose a JLRO iteration algorithm to solve the original problem based on the block coordinate descent theory.

### 3.2. UAV 3D Location Subproblem

With a given resource allocation scheme, problem (8) is simplified into the following problem:
(9a)$$\begin{array}{cc} \; & \max_{\theta ,\pi ,x_{U},y_{U},z_{U}}\pi \end{array}$$
(9b)$$\begin{array}{cc} s.t.\;\;\; & \left(6b),\;(6c),\;(6d),\;(7c),\;(8c),\;(8d\right) \end{array}$$

The above problem (9) is nonconvex due to nonconvex constraints (7c) and (8c).

Constraint (7c) is further equivalently divided into the following two constraints using slack variable Γ:(10)$$\pi \leq \frac{P_{s}\eta \phi 10^{-\frac{L_{a}}{10}}10^{-\frac{L_{h}}{10}}A_{R}}{\Gamma E_{p}N_{b}}$$
(11)$$\Gamma \geq 0.25\pi \psi^{2}d_{S,R}^{2}$$

Note that constraint (Equation 11) is convex with respect to the UAV location, but constraint (Equation 10) is a nonconvex constraint because $\frac{1}{\Gamma }$ in right side of constraint (Equation 10) is not concave with respect to Γ.

However, we can observe that $\frac{1}{\Gamma }$ is a convex function of Γ and can thus be handled using the successive convex approximation method. For a given reference input $\Gamma_{\epsilon }$ in the ϵth iteration of the successive convex approximation, the concave lower bound of $\frac{1}{\Gamma }$ can be obtained through the first order Taylor expansion:(12)$$\frac{1}{\Gamma }\geq \frac{1}{\Gamma_{\epsilon }}-\frac{\Gamma -\Gamma_{\epsilon }}{\Gamma_{\epsilon }^{2}}=\frac{2\Gamma_{\epsilon }-\Gamma }{\Gamma_{\epsilon }^{2}}=f_{\frac{1}{\Gamma }}^{low}$$

The nonconvex property of constraint (8c) is caused by the complex through-wall fading model. To cope with this difficulty, constraint (8c) is further divided into the following three constraints with the aid of slack variables $\omega_{i}$ and $\gamma_{i}$:(13)$$\theta \leq B_{i}\log_{2}\left(1+\frac{P_{i}\omega_{i}}{B_{i}N_{0}}\right),\forall i$$
(14)$$\omega_{i}\leq 10^{-\frac{PL_{Fi}+\beta_{1}+\beta_{2}{\left(1-\gamma_{i}\right)}^{2}+PL_{Ii}}{10}},\forall i$$
(15)$$\gamma_{i}\leq \frac{x_{U}-x_{B}}{d_{D_{i},R}},\forall i$$

Now, we can find that constraint (Equation 13) is convex, but constraints (Equation 14) and (Equation 15) need further transformation.

Constraint (Equation 14) can be equivalently changed as follows:(16)$$\ln\omega_{i}+\frac{\ln10}{10}\left(PL_{Fi}+\beta_{1}+\beta_{2}{\left(1-\gamma_{i}\right)}^{2}+PL_{Ii}\right)\leq 0,\forall i$$

In the left side of constraint (Equation 16), $\ln\omega_{i}$ and $PL_{Fi}$ are concave, and thus when using the first order Taylor expansion, the following two convex upper bounds are given:(17)$$\ln\omega_{i}\leq \ln\omega_{i|\epsilon }+\frac{\omega_{i}-\omega_{i|\epsilon }}{\omega_{i|\epsilon }}=f_{\ln\omega_{i}}^{upper},\forall i$$
(18)$$\begin{array}{cc} PL_{Fi} & \leq 20\log_{10}\left(\frac{4\pi fd_{D_{i},R|\epsilon }}{c}\right)\\ \; & +\frac{20(d_{D_{i},R}-d_{D_{i},R|\epsilon })}{d_{D_{i},R|\epsilon }\ln10}=f_{PL_{Fi}}^{upper},\forall i \end{array}$$
where $d_{D_{i},R|\epsilon }=\sqrt{{\left(x_{U|\epsilon }-x_{B}\right)}^{2}+{\left(y_{U|\epsilon }-y_{i}\right)}^{2}+{\left(z_{U|\epsilon }-z_{i}\right)}^{2}}$ with $\left(x_{U|\epsilon },y_{U|\epsilon },z_{U|\epsilon }\right)$ as the ϵth UAV location input in the successive convex approximation, and $\omega_{i|\epsilon }$ is the ϵth input in the successive convex approximation.

Using a similar method, we can change constraint (Equation 15) as follows:(19)$$\ln\gamma_{i}+\ln d_{D_{i},R}\leq \ln\left(x_{U}-x_{B}\right),\forall i$$

To save space, we only define $f_{\ln\gamma_{i}}^{upper}$ and $f_{\ln d_{D_{i},R}}^{upper}$ as the corresponding convex upper bounds of $\ln\gamma_{i}$ and $\ln d_{D_{i},R}$, because they have similar form as that used in (Equation 17).

In summary, by dealing with nonconvex constraints (7c) and (8c), problem (9) is changed for solving the following convex problem when performing the ϵth iteration of the successive convex approximation:
(20a)$$\begin{array}{cc} \; & \max_{\theta ,\pi ,\Gamma ,\omega_{i},\gamma_{i},x_{U},y_{U},z_{U}}\pi \end{array}$$
(20b)$$\begin{array}{cc} s.t.\;\;\; & \left(6b),\;(6c),\;(6d),\;(8d),\;\;(11),\;\;(13\right) \end{array}$$
(20c)$$\begin{array}{cc} \; & \pi \leq \frac{P_{s}\eta \phi 10^{-\frac{L_{a}}{10}}10^{-\frac{L_{h}}{10}}A_{R}f_{\frac{1}{\Gamma }}^{low}}{E_{p}N_{b}}\\ \; & f_{\ln\omega_{i}}^{upper}+\frac{\ln10}{10}(f_{PL_{Fi}}^{upper}+\beta_{1} \end{array}$$
(20d)$$\begin{array}{cc} \; & +\beta_{2}{\left(1-\gamma_{i}\right)}^{2}+PL_{Ii})\leq 0,\forall i \end{array}$$
(20e)$$\begin{array}{cc} \; & f_{\ln\gamma_{i}}^{upper}+f_{\ln d_{D_{i},R}}^{upper}\leq \ln\left(x_{U}-x_{B}\right),\forall i \end{array}$$

The current form of problem (20) is that of a standard convex, and many approaches, such as interior point method, can be applied to solve this problem.

### 3.3. Resource Allocation Subproblem

With a given UAV location, problem (8) is changed into the following form:
(21a)$$\begin{array}{cc} \; & \max_{\theta ,\pi ,B_{i},P_{i}}\pi \end{array}$$
(21b)$$\begin{array}{cc} s.t.\;\;\; & \left(6e),\;(6f),\;(6g),\;(7c),\;(8c),\;(8d\right) \end{array}$$

Only constraint (8c) is nonconvex in the above problem (21).

To handle the nonconvex constraint (8c), it is further divided into the following constraints with the aid of $\Omega_{i}$ and $\Phi_{i}$:(22)$$\begin{array}{c} \theta \leq B_{i}\Omega_{i},\forall i \end{array}$$
(23)$$\begin{array}{c} \Omega_{i}\leq \log_{2}\left(1+\frac{\Phi_{i}10^{-\frac{PL_{i}}{10}}}{N_{0}}\right),\forall i \end{array}$$
(24)$$\begin{array}{c} \Phi_{i}\leq \frac{P_{i}}{B_{i}},\forall i \end{array}$$

Constraint (Equation 23) satisfies the convex rule. Constraints (Equation 22) and (Equation 24) are nonconvex constraints and are observed to be similar in form. Taking constraint (Equation 22) as an example, it can be transformed into:(25)$$\begin{array}{c} \ln\theta \leq \ln B_{i}+\ln\Omega_{i},\forall i \end{array}$$

The convex upper bound of lnθ has the same form as $\ln\omega_{i}$ in (Equation 17) and is, thus, also omitted. In general, we define $f_{\ln\theta }^{upper}$, $f_{\ln\Phi_{i}}^{upper}$, and $f_{\ln B_{i}}^{upper}$ as the corresponding convex upper bounds of lnθ, $\ln\Phi_{i}$, and $\ln B_{i}$, respectively.

Therefore, in the ϵth iteration of the successive convex approximation, problem (21) is changed into the following convex problem for solving:
(26a)$$\begin{array}{cc} \; & \max_{\theta ,\pi ,\Omega_{i},\Phi_{i},B_{i},P_{i}}\pi \end{array}$$
(26b)$$\begin{array}{cc} s.t.\;\;\; & \left(6e),\;(6f),\;(6g),\;(7c),\;(8d),\;(23\right) \end{array}$$
(26c)$$\begin{array}{cc} \; & f_{\ln\theta }^{upper}\leq \ln B_{i}+\ln\Omega_{i},\forall i \end{array}$$
(26d)$$\begin{array}{cc} \; & f_{\ln\Phi_{i}}^{upper}+f_{\ln B_{i}}^{upper}\leq \ln P_{i},\forall i \end{array}$$

### 3.4. Proposed JLRO Algorithm

In order to solve the original problem (8), the block coordinate descent theory can be applied, such that at each iteration, only two standard convex optimization problems (20) and (26) are alternatively solved, as shown in Algorithm 1.    
**Algorithm 1:** JLRO algorithm1. Set ϵ=0. Initialize $P_{i|0}$, ${\mathrm{var}}_{20}^{0}=\{x_{U|0},y_{U|0},z_{U|0},\Gamma_{0},$   $\gamma_{i|0}$} and ${\mathrm{var}}_{26}^{0}=\left\{\theta_{0},\Phi_{i|0},B_{i|0}\right\}$;2. Repeat3.   Solve problem (20) with given $P_{i|\epsilon }$ and ${\mathrm{var}}_{20}^{\epsilon }$, and     denote the solution as ${\mathrm{var}}_{20}^{\epsilon +1}$;4.   Solve problem (26) with given $x_{U|\epsilon +1}$, $y_{U|\epsilon +1}$ and     $z_{U|\epsilon +1}$, and denote the solution as $P_{i|\epsilon +1}$ and ${\mathrm{var}}_{26}^{\epsilon +1}$;5. $|{\mathrm{var}}_{26}^{\epsilon +1}-{\mathrm{var}}_{20}^{\epsilon +1}|>\xi$. Update ϵ=ϵ+1;6. Until7. Converge to a predefined tolerant error.

In step 1 of Algorithm 1, $P_{i|0}$ needs to be initialized because problem (20) is formulated under given resource allocation. ${\mathrm{var}}_{20}^{0}$ and ${\mathrm{var}}_{26}^{0}$ are initialized because of cyclic iteration in the successive convex approximation. Specifically, $P_{i|0}$ is initialized as $P_{i|0}=\frac{P_{max}}{n}$ according to constraint (6g); $B_{i|0}$ is initialized as $B_{i|0}=\frac{B_{max}}{n}$ according to (6f); the initial UAV location is set as $\left(x_{U|0}=\frac{x_{U}+x_{S}}{2},y_{U|0}=\frac{y_{B}}{2},z_{U|0}=\frac{z_{B}}{2}\right)$; $\Gamma_{0}$ and $\gamma_{i|0}$ are initialized based on strict equality in constraints (Equation 11) and (Equation 15), respectively; $\theta_{0}$ and $\Phi_{i|0}$ are initialized based on strict equality by jointly using constraints (Equation 22)–(Equation 24).

As can be seen from the proposed JLRO algorithm, at each iteration the JLRO algorithm only solves two basic convex problems with each having a polynomial computational complexity. Thus, the total complexity of the proposed algorithm is $O(N_{c}*N_{var}^{3.5})$, where $N_{c}$ is the number for convergence and $N_{var}$ is the total number of variables at each iteration.

## 4. Simulation Results

The simulation parameters used in this paper are summarized in Table 2. Unless otherwise stated, all simulations and experiments assume that the number of indoor users is 10. The location coordinates of 10 indoor users are shown in Table 3. The user coordinates in Table 3 are taken to be the same as those in reference [5] in order to facilitate subsequent comparative experiments. Unless otherwise specified, the simulation experiments in this paper are carried out according to the above parameters.

Figure 3 shows the convergence of the JLRO algorithm, where “60In”, “50In”, “40In”, “30In”, “20In”, and “10In” indicate that the number of indoor users is 60, 50, 40, 30, 20, and 10, respectively. All results are the average of five random values. For example, $30In=\frac{A_{1}\sum_{i=1}^{30}T_{i}+A_{2}\sum_{j=1}^{30}T_{j}+A_{3}\sum_{c=1}^{30}T_{c}+A_{4}\sum_{v=1}^{30}T_{v}+A_{5}\sum_{m=1}^{30}T_{m}}{5}$. The values of A1, A2, A3, A4, and A5 are all 1, which means that 30 user coordinates are randomly generated each time. $\sum_{i=1}^{30}T_{i}$ represents the sum of the throughput of 30 users. The experimental results show that the algorithm can converge when the number of iterations is 8. This shows that the number of indoor users will only affect the final convergence value and will not destroy the convergence characteristics of the algorithm. The simulation results also show that the total throughput increases with the increase in the number of indoor users, but this trend is becomes increasingly diminished. In this simulation, when the number of users increases from 50 to 60, the throughput still increases, but the increase is small. When the number of users increases to about 63, the throughput basically remains unchanged. However, it is not found that the throughput will decline.This is because in the case of limited resources, the increase in the number of users will improve the utilization of resources, but once a certain threshold is exceeded, it will not be able to provide communication services for more users. This threshold must exist, although it is related to the location coordinates of all users and, thus, varies with changes in these coordinates. When sufficient coordinates are input, the result of the JLRO algorithm can be calculated. In reality, however, the initial threshold is generally set according to the upper limit of the number of air ports. After processing the JLRO algorithm, a new threshold is calculated, and the final threshold is that which is lowest of the new threshold and the initial threshold.

Figure 4 shows the bandwidth allocation and power allocation in the JLRO algorithm. The number of indoor users is 10. The following conclusions can be drawn from the figure: 1. The sum of all sub bandwidths and power are equal to $B_{max}$ and $P_{max}$. In other words, bandwidth and power resources have been fully utilized. 2. With the increase in $B_{max}$ and $P_{max}$, the sub bandwidth and power increases. 3. The increase in sub bandwidth and sub power is not linear. This is because bandwidth and power are convolved together and affect each other. 4. Compared with bandwidth allocation, power allocation is more regular. In other words, although the increase in sub power is not linear, it is nearly linear. In the description of Figure 5, *R* is the coordinate parameters of UAV and R=(50.55,34.32,58.96). In the three coordinates of D5, D9, and D10, due to the influence of ordinate, the distance between D10 and UAV is the shortest and the loss is the smallest. The distance between D5 and D9 and UAV is relatively far, and the loss is relatively large. The JLRO algorithm takes the principle of user fairness into account when allocating power and bandwidth, so the bandwidth and power allocated for D10 is relatively small, while the bandwidth and power allocated for D9 and D5 are relatively large. In addition, bandwidth and power allocation are convolved and difficult to analyze separately.

Figure 6 shows the throughput affected by $B_{max}$. Figure 7 shows the throughput affected by $P_{max}$. The number of indoor users is 10. The conclusions are as follows: 1. With the increase in $B_{max}$ and $P_{max}$, the throughput increases significantly. 2. As can be seen from Figure 7, with the increase in $B_{max}$, the throughput seems to increase linearly. This is, however, a visual error caused by pixels, and the increase in throughput is nonlinear. This is because the noise power is also increasing with the bandwidth, which causes a decline in throughput. 3. From the perspective of slope, the throughput increases more rapidly when the increase in bandwidth of the same multiple is relatively higher than the power increase. 4. From the value of throughput, increasing the bandwidth of the same multiple will result in increased throughput. 5. The premise of these conclusions is $\sum_{i=1}^{n}T_{i}\leq C$. Otherwise, increasing bandwidth and power is ineffective.

Figure 8 shows the comparison of throughput between FSO/RF and RF. The object of comparison is RF from the literature [5]. In Figure 9, the parameter used by FSO/RF is $P_{max}=0.1w$, and the parameter used by RF is $P_{1}=0.1w,P_{2}=0.5w$, where $P_{1}=0.1w$ and $P_{2}=0.5w$ represent the transmission power of the first hop and the second hop, respectively. The simulation results show that the throughput using FSO/RF is about twice that using RF alone. With the increase in bandwidth, this gap will become larger and larger. The reason is that in the communication system proposed in reference [5], a large part of the bandwidth should be allocated to the first hop link. The first hop in this paper is FSO, which does not occupy this part of bandwidth resources. All the bandwidth in this paper is used for the second hop link.

Figure 9 shows the fairness between links. The abscissa represents the label of each link. The simulation results show that the JLRO algorithm proposed in this paper can ensure the fairness between links; that is, the transmission rate of each two links is the same. The blue column indicates the link transmission rate when fairness is not considered. It can be seen that the transmission rate of link “2.3.5.7.8.9” is a. A is the transmission rate set in advance to meet the minimum communication needs. The transmission rate of “1.4.6.10” is significantly higher, especially link “10”. This is because when fairness is not considered, the system will give priority to allocating bandwidth and power to the link with the best revenue, so as to maximize the total throughput of the system. With the increase in power and bandwidth, the income of the link will become lower and lower. At this time, the system will allocate bandwidth and power to the link with higher income. This cycle continues until the bandwidth and power are fully allocated. The sum of the blue columns must therefore greater than that of the red columns. This demonstrates that fairness comes at the expense of partial throughput.

Figure 5 shows the deployment location of the UAV. D represents indoor users in Table 3. R represents the UAV deployment position calculated by JLRO. The coordinate parameters of UAV is (50.55,34.32,58.96). The simulation results show that the abscissa of the UAV is very close to the specific wall relative to the distance between the user and the outdoor base station. The Y and Z coordinates of UAV are greatly affected by user coordinates. If there are more users, the UAV may be closer to the center of the wall. In addition, JLRO algorithm can dynamically determine the location of UAV according to the user’s movement to ensure the user’s communication.

## 5. Conclusions

In this paper, FSO/RF is introduced into an outdoor–indoor communication scenario. Based on the wireless optical communication model, an outdoor–indoor communication model is established. By decomposing the original problem, a fast convergent JLRO algorithm is proposed. User position coordinates do not affect the performance of JLRO algorithm. The algorithm jointly optimizes the deployment location and resource allocation of UAV and maximizes the throughput of communication system on the premise of ensuring user fairness and fully considering information causality. 

## Figures and Tables

**Figure 1 sensors-23-02541-f001:**
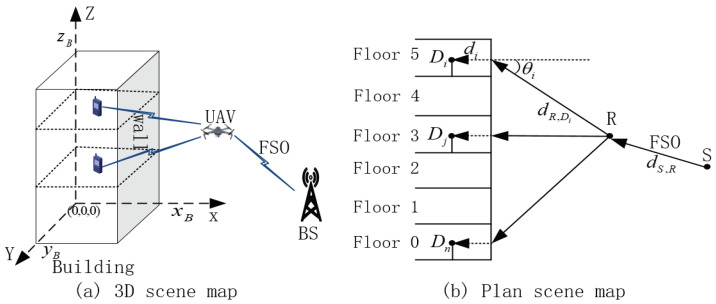
UAV relay provides long distance communication service for indoor users.

**Figure 2 sensors-23-02541-f002:**
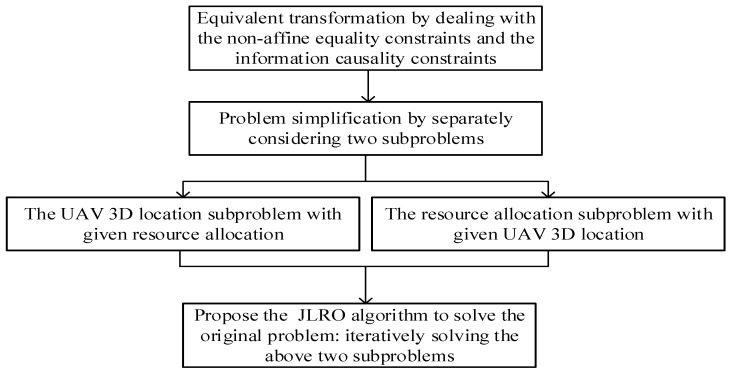
Flow chart for solving the formulated problem.

**Figure 3 sensors-23-02541-f003:**
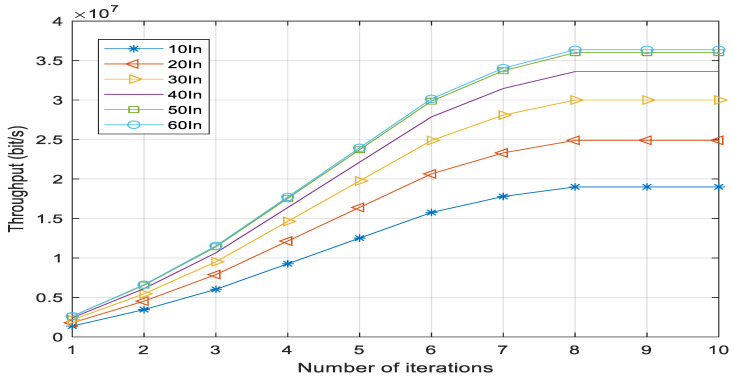
JLRO algorithm convergence.

**Figure 4 sensors-23-02541-f004:**
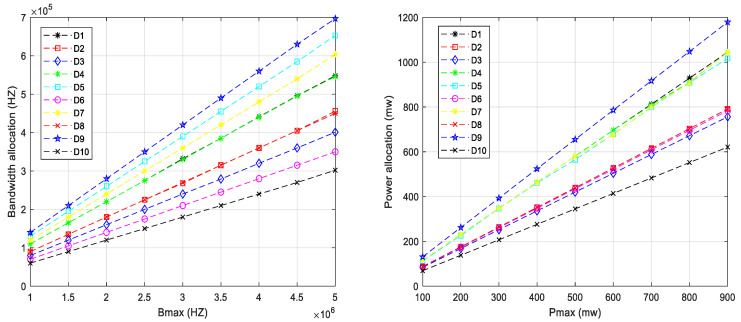
Bandwidth allocation and power allocation in the JLRO algorithm.

**Figure 5 sensors-23-02541-f005:**
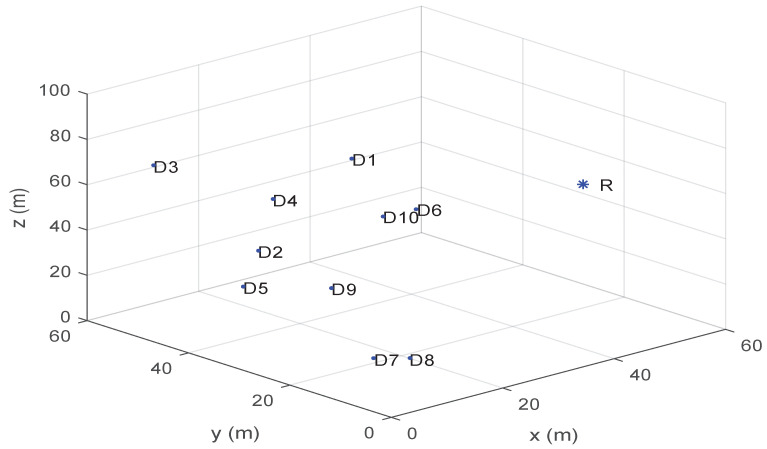
Deployment location of UAV.

**Figure 6 sensors-23-02541-f006:**
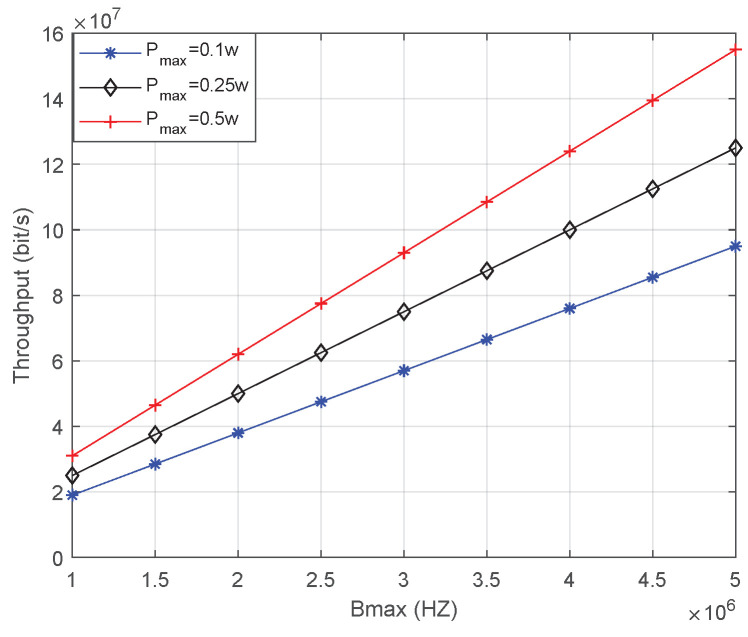
Throughput is affected by $B_{max}$.

**Figure 7 sensors-23-02541-f007:**
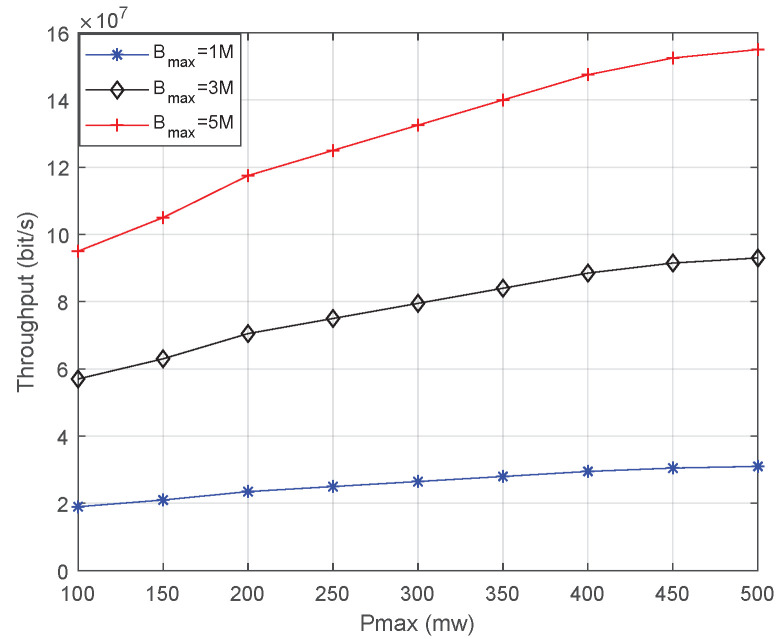
Throughput is affected by $P_{max}$.

**Figure 8 sensors-23-02541-f008:**
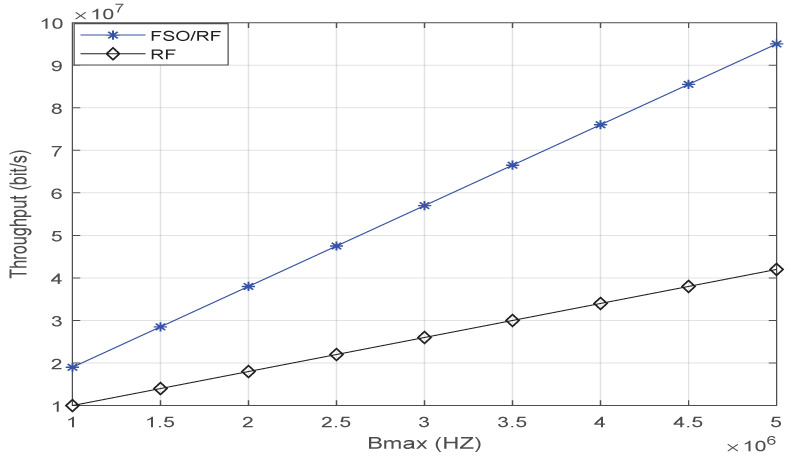
Comparison of throughput between FSO/RF and RF.

**Figure 9 sensors-23-02541-f009:**
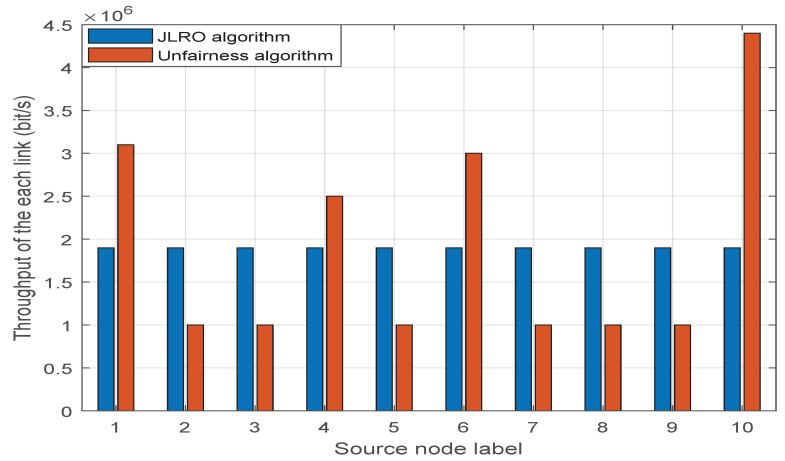
Link fairness.

**Table 1 sensors-23-02541-t001:** Summary of notations.

Notation	Description
*S*	Source node
*D*	Destination node
*R*	UAV relay
$\left(x_{•},y_{•},z_{•},\right)$	Coordinates of •
*d*	Distance between * to *
*f*	Carrier frequency
$\theta_{i}$	Elevation angle
*C*	Transmission rate of the first step
*T*	Transmission rate of the second step
$E_{p}$	Photon energy
$L_{g}$	The geometrical loss
$L_{a}$	Atmospheric attenuation
$h_{p}$	Planck’s constant
$P_{s}$	Transmission power of source node
ϕ	The optical efficiency
$N_{b}$	Sensitivity of UAV receiving equipment
PL	The path loss
$P_{max}$	Maximum transmitting power of UAV
$B_{max}$	Maximum transmitting bandwidth of UAV
$P_{i}$	Power allocated to the *i*-th user
$B_{i}$	Bandwidth allocated to the *i*-th user

**Table 2 sensors-23-02541-t002:** Summary of simulation parameters.

Parameter	Value	Parameter	Value
$x_{B}$	20 m	$y_{B}$	50 m
$z_{B}$	100 m	$\left(x_{S},\;y_{S},\;z_{S}\right)$	(1000, 25, 50)
$P_{S}$	200 mWatt	ηϕ	$e^{0.2}$
$L_{h}$	2 dB	ψ	1 mrad
*r*	0.04 m	λ	1550 nm
$N_{b}$	100 photons/b	$h_{p}$	$6.626\times 10^{-34}$ J-S
*c*	$3\times 10^{8}$ m/s	*f*	1 GHz
$N_{0}$	−174 dbm/Hz	$B_{\max}$	1 MHz
$P_{\max}$	100 mWatt	ξ	$10^{-5}$

**Table 3 sensors-23-02541-t003:** The coordinates of indoor users.

$D_{1}$	$D_{2}$	$D_{3}$	$D_{4}$	$D_{5}$
(12,21,91.5)	(8,35,43.5)	(1,48,76.5)	(17,42,55.5)	(8,38,25.5)
$D_{6}$	$D_{7}$	$D_{8}$	$D_{9}$	$D_{10}$
(9,5,82.5)	(5,9,16.5)	(17,15,4.5)	(12,25,31.5)	(14,17,67.5)

## Data Availability

Data sharing is not applicable.

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
