# Peer review of "A Location-Aware Resource Optimization for Maximizing Throughput of Emergency Outdoor–Indoor UAV Communication with FSO/RF"

_sensors, 2023, doi:10.3390/s23052541_

Round 1
Reviewer 1 Report
The organization of the article and contribution is sufficient to be published in this journal; however, there are some changes that need to be done before the publication.
1. The main core of the article must be aligned with the work done and the title of the article should be modified accordingly, i.e., recheck the problem formulation and modify the title.
2. The 3D localization of users is not analyzed and compared in this paper.
3. Section 4, simulation results: All simulations and experiments assume that the number of indoor users is 10. However, in Figure 3, the simulation is performed with 30 indoor users as well. It is suggested to recheck the statement.
4. The position of 10 indoor users is assumed to be fixed, which is not a legitimate assumption. It is suggested to include a random position of indoor users for each simulation and include a generalized/averaged simulation results.
5. In Figure 3, the incorporation of 30 indoor users provide a better throughput; therefore, it is suggested to check and verify the maximum number of indoor users for specific scenario. It seems that the inclusion of 40 indoor users may provide better throughput.
6. Figures 4 and 5, and their explanation looks very similar, in fact, they are. Therefore, it is better to merge the details of both figures and write concisely. The main purpose of these figures is not clear.
7. In Figures 4 and 5, different simulation results will come for bandwidth and power allocation. If the same results are considered, then the authors should clarify that why D9 is allocated with the highest bandwidth (power), while D10 is allocated with lower bandwidth (power), when they are close to each other compared to D5.
8. In the last, the authors should go through the whole manuscript for English language correction.
Author Response
All of responses to the Reviewer’comments are in the word letter

Reviewer 2 Report
How the coordinates of indoor users D1 to D10 are decided. It is not clarified
The wireless optical communication model is not verified and needs to be clear in conclusion with the detailed outcome
In the entire manuscript, authors have used we have done this .. we did this, the manuscript should be written with respect to the third party
Power allocation in the JLRO algorithm, throughput and convergence and bandwidth results need to discuss in more detail so that the results verifications can be done
Compare the work with existing if possible
Author Response

(The authors gave the same response as above.)

Reviewer 3 Report
The topic of this paper is interesting and timely but the reviewer has several concerns,
· The reviewer suggest to explain at the beginning of the system model (and eve in the abstract) that FSO is only used from BS to UAV. When we start to read the paper we be the impression that you are also using FSO from UAV to users that do not make sense.
· What frequency band are you assumed from UAV to indoor users?
· Please improve the quality of the flow chart.
· The complexity of the proposed optimization algorithms is not provided. This is important to understand its feasibility for practical implementations.
· In the simulation section some parameters used to obtain the results are missing. What frequencies were used from BS-UAV and from UAV-Indoor user? What channel models were assumed in these two different links?
· The quality of the results figures is not good, namely the legends.
· Why the results are not compares with other approaches?
Author Response

(The authors gave the same response as above.)

Round 2
Reviewer 1 Report
1. The authors have modified the title of the article; however, the title still lacks to provide the exact sense of the work done in the paper. The title should be modified as follows.
“A Location-aware Resource Optimization for Maximizing Throughput of Emergency Outdoor-Indoor UAV Communication with FSO/RF”
2. The proposed work is mainly dependent on the locations of the users within the building, and the user locations are generated randomly to check the performance of the proposed algorithm; however, the authors are not clear why the location of the users is important to their work. The changes of users’
locations would impact because it will suffer the bandwidth and power allocation. If it is assumed that the proposed algorithm is not linked with the locations of the users then the response of authors is acceptable. In the current scenario, the performance of algorithm must be explored. For instance, what
would be the impact of the algorithm when all the users are located at the same floor and close to each other. In my opinion, the algorithm inherently adjusts and maximizes the throughput in a specific scenario despite the locations of the users. In that case, it would be a simple throughput maximization instead of location-based or 3D location based throughput maximization. The authors should also recheck their statement written in the conclusion “The algorithm jointly optimizes the deployment location”. What is the purpose of this statement, please elaborate.
Author Response
All of responses to the Reviewer’ comments are in the word letter

Reviewer 2 Report
The corrections and clarifications are satisfactory
Author Response
Reviewer expressed his satisfaction with the responses
Reviewer 3 Report
The reviewer main concerns have been satisfactory addressed.
Author Response

(The authors gave the same response as above.)
